# Excess Mortality of Males Due to Chronic Obstructive Pulmonary Disease (COPD) in Poland

**DOI:** 10.3390/healthcare12040437

**Published:** 2024-02-08

**Authors:** Waclaw Moryson, Barbara Stawińska-Witoszyńska

**Affiliations:** Department of Epidemiology and Hygiene, Chair of Social Medicine, Poznan University of Medical Sciences, Rokietnicka 4, 60-806 Poznan, Poland; bwitoszynska@ump.edu.pl

**Keywords:** excess mortality of males, COPD, epidemiology

## Abstract

At present, female life expectancy exceeds male life expectancy almost worldwide. However, numerous studies indicate that this disparity is gradually decreasing. In Poland, the gender gap in life expectancy peaked in 1991 when it amounted to 9.2 years. Since then, a narrowing of the gap has been observed, reaching 8 years in 2021. Decreasing differences in life expectancy between men and women in Poland were mainly the result of a reduction in mortality due to ischaemic heart disease, cerebrovascular disease, and a number of malignancies.Less attention has been paid to chronic obstructive pulmonary disease (COPD) although it is the third leading cause of death worldwide. This paper includes an analysis of mortality due to chronic obstructive pulmonary disease COPD. The male excess mortality was calculated as the ratio of mortality rates in the male population scaled up to the corresponding rates in the female population using both crude and standardised detailed mortality rates. The Joinpoint model was used to determine time trends. It was shown that from 2008 to 2021, the excess mortality of men due to COPD in Poland decreased by 3.3% per year from 2.4 to 1.7 when using crude coefficients, while when standardised coefficients were applied, it decreased significantly by 3.9% per year from 3.8 to 2.4. The decrease in the excess mortality of men in Poland was due to a simultaneous decrease in mortality in the population in general; however, a greater decrease was observed in the male population. The mortality of men and women, and, at the same time, the excess mortality of men caused by COPD in Poland decreased faster in the period studied than in other European countries.

## 1. Introduction

This study presents an epidemiological analysis of decline inthe excess mortality of males due to chronic obstructive pulmonary disease (COPD) in Poland between 2008 and 2021. The analysed period includes years of positive changes in the lifestyle of Poles and of improvement in the quality of healthcare, but also a time of significant changes in the age structure of the Polish population.

Since the end of the 19th century, female life expectancy has exceeded that of males in almost all countries in the world. However, at the very beginning of the 20th century, the difference was of only two years. Initially, the main factor contributing to the longer life expectancy of women was the reduction in birth-related mortality. In the following decades, an increasing disproportion in life expectancy in favour of women continued to widen, especially after the end of the Second World War [1]. Higher male mortality in younger age groups was due to a higher incidence of injury and suicide, and, in older age groups, due to behavioural changes affecting the development of chronic diseases and increased mortality from cardiovascular disease and cancer [2]. Historically, women are less likely to smoke cigarettes or drink alcohol, and they follow a healthier diet, which means they are less likely to be exposed to risk factors for developing the most common lifestyle diseases [3,4]. Currently, however, several studies point to a gradual narrowing of the gender gap in life expectancy, which, in some developed countries, has already been occurring since the 1970s [5,6,7].

In Poland, the difference in life expectancy between men and women peaked in 1991, with a value of 9.2 years. Since then, a narrowing of the gap has been observed, decreasing to 8 years in 2021. Life expectancy for males increased between 1990 and 2021 from 66.2 years to 71.8 and for females from 75.2 years to 79.7 years [8]. However, the 2021 data highlighted a reduction in the average life expectancy of Poles in 2021 compared to 2019 as a result of the COVID-19 pandemic [9].

The narrowing of the gap in male and female life expectancy in Poland was mainly due to a reduction in mortality due to ischaemic heart disease, cerebrovascular disease and a number of malignancies, which was still very high in the 1990s. In the area of cardiovascular diseases, a simultaneous decrease in mortality in men and women was observed, yet its rate was noticeably higher in the male population. Standardised mortality rates for women aged 25–64 in Poland decreased between 1990 and 2002 by 47.4 per 100,000 (the largest decrease among all European countries), while the value of standardised rates for the male population decreased by 117.5 per 100,000. Mortality due to cancer in the male population started to decrease at the beginning of the 21st century, while in the female population, a relatively stable level of mortality due to cancer was observed [10,11,12]. In contrast, less attention is paid to chronic obstructive pulmonary disease (COPD) despite its key relevance in terms of public health. In Poland, COPD is the fifth most common cause of mortality, and globally, the third most common cause of death [13].

This study investigates mortality due to COPD in the Polish population. The number of COPD cases is changing with preventive measures to reduce smoking or air pollution, among other things. Incidentally, the same factors have contributed to changes in mortality from cardiovascular disease and cancer [14]. Previous studies on COPD demonstrated that in recent years, a significant reduction in mortality from COPD had been observed in Poland, particularly in the male population, who are more likely to suffer and die from COPD than females [15].

## 2. Materials and Methods

The present study analysed all registered deaths in the Polish population in 2008–2021. The data on population size and number of deaths were obtained from the Central Statistical Office. The deaths analysed were those in which the underlying cause was registered with codes J40–J44 according to the International Statistical Classification of Diseases and Related Health Problems (ICD-10) [16].

Crude and standardised mortality rates per 100,000 people were used to analyse the mortality phenomenon for both sexes. The standardisation of the coefficients was carried out to eliminate the effect of differences in age structure on the intensity of deaths based on the standard European population in 2013. The magnitude of the male excess mortality phenomenon was calculated as the ratio of the male population mortality rates to the corresponding rates in the female population using both crude and standardised detailed mortality rates.

The Joinpoint model was used to analyse time trends in male and female mortality and male excess mortality. The Joinpoint model is a special version of linear regression where the time trends are represented by a broken curve in which the segments describing the dynamics of a change in the trend in a certain time interval join at the points where the change in the time trend is significant. In the analysis carried out, a minimum number of trend change points of zero and a maximum number of one were assumed. The significance of the model containing a change point in the trend dynamics was verified using the Monte Carlo permutation method.The time trend analysis using the Joinpoint model rendered it possible to determine the average annual percentage change (AAPC) in mortality and excess mortality in men due to COPD. If the year in which the change in trend dynamics occurred was determined within the time interval studied, the annual percentage change (APC) was also determined both before and after the year in which the change occurred. The analysis was performed using the Joinpoint Regression statistical software version 4.7.0.0 (US National Cancer Institute, Bethesda, MD, USA). For both the annual percentage change in APC and the mean annual percentage change in the rates, a 95% confidence interval of −95% CI was determined, with *p* < 0.05 assumed as the level of significance.

The mortality assessment carried out applied to the entire population of men and women, although mortality rates increase with age and the majority of deaths from COPD in Poland occur in people over 50 years of age [17].

## 3. Results

This study points to the fact that, in Poland, mortality due to COPD in 2008–2021 was higher in the male population than in the female population. Yet, the gap significantly narrowed. Between 2008 and 2021, the total reduction in male excess mortality was 36.1% for male excess mortality described by standardised rates, and 31.7% for male excess mortality described by crude rates. The decrease in male excess mortality was due to a simultaneous decrease in male and female mortality rates, though a more significant decline was observed in the male population.

The time trend analysis of male mortality due to COPD showed a significant decrease in crude rates over the study period by 3.4% per year. While the crude mortality rate for men was 23.4 per 100,000 in 2008, in 2021, it was only 16.9 per 100,000. The reduction in standardised rates was even more pronounced, amounting to 5.6% per year over the entire period analysed, and characterised by statistical significance, a decrease from 52.2 per 100,000 to 22.4 per 100,000. The figures of female crude mortality rates due to COPD did not change significantly between 2008 and 2021. However, a significant decrease in standardised rates of 1.8% per year was observed, and the value of this rate decreased from 13.7 per 100,000 in 2008 to 9.2 in 2021. The Joinpoint analysis conducted did not show any change in the dynamics of the reduction trend, both in crude and standardised mortality rates for men and women, during the 2008–2021 period (Table 1).

Of note is the difference in the values of standardised and crude mortality rates. Standardised mortality rates are higher than crude rates in both the female and male populations, although the difference is more pronounced in the male population. In 2008, the standardised rate value was 13.7 per 100,000 in the female population, which was 15% higher than the crude rate value (12 per 100,000). In subsequent years, the gap gradually narrowed until 2018, when the values of standardised rates in the female population were already smaller than the crude rates. In 2021, the standardised mortality rate for women was 9.2 per 100,000, 9% lower than the value of its crude equivalent (10.1 per 100,000). In the male population in 2008, the value of the standardised rate was 52.2 per 100,000, 78% higher than the value of the crude rate (29.4 per 100,000). Both in the male population and in the female population, the difference between the values of the standardised mortality rates and their crude equivalents decreased from year to year. However, among men, the values of the standardised rates were higher than the crude ones in the study period. In 2021, the standardised mortality rate for men was 22.4 per 100,000, 22% higher than the crude rate (16.9 per 100,000). The large difference between the values of the crude and standardised rates meant that the values of male excess mortality rates also varied greatly depending on whether crude or standardised mortality rates were used to determine them. While the male excess mortality rate determined using crude rates was 2.45 in 2008, the male excess mortality rate determined using standardised rates was 3.81.

The analysis of time trends in male excess mortality due to chronic obstructive pulmonary disease showed a decrease in this phenomenon over the entire period from 2008 to 2021. As in the case of the analysis of mortality rates, the Joinpoint analysis did not show any change in the dynamics of the trend of reduction in male excess mortality over the study period when the crude and standardised rates were considered. Male excess mortality, in the case of crude rates, decreased significantly by 3.3% per year between 2008 and 2021 from a level of 2.4 to a level of 1.7. Using standardised rates, it decreased significantly by 3.9% per year from 3.8 to 2.4 (Figure 1).

## 4. Discussion

### 4.1. Principal Findings

Mortality due to COPD tends to be more prevalent in the male population, although the period 2008–2021 saw a significant reduction in this disparity in Poland. Male excess mortality in Poland attributable to COPD decreased as a result of a greater decline in male mortality than in female mortality. A difference in the rate of reduction in both standardised and crude coefficient values in favour of men was noted. These changes occurred despite lower exposure to tobacco in the female population and a concomitant decrease in exposure to tobacco smoke in both sexes. The proportion of smokers in Poland between 2009 and 2019 decreased from 35% to 24% in the male population and from 24% to 18% in the female population (Table 2). While for the past decades, nicotinismhas beenfar more prevalent among men than women, this difference in Poland has been gradually disappearing in recent years. In 2019, in Poland, the percentage of people who started smoking regularly in the past 12 months was higher in the female population (12%) than in the male population (10%) [18]. The reduction in excess mortality observed in men may be attributed to the fact that females develop COPD with acumulative exposure to tobacco smoke lower than that of males. At the same time, the female population has a higher non-tobacco-related incidence of COPD. The course of the disease in the female population is also associated with a more rapid deterioration of lung function [19,20], among others. During adolescence, the growth of the airways in women is slight compared to the growth of the lungs in men, in whom the bronchopulmonary growth is more homogeneous, andthusinhaled substances adhere to a smaller surface area. In healthy women, there is greater deposition of particles than in men. In addition, tobacco smoke has an anti-oestrogenic effect, which further impairs the maintenance of lung elasticity and airway patency [21].

This study revealed differences in male excess mortality depending on the rates used. Male excess mortality calculated as a quotient of crude mortality rates is lower, which is due to the different age structure of the male and female populations in Poland. Standardisation makes it possible to compare the values of rates, here mortality rates, between populations that differ in age structure. While crude mortality rates reflect the actual magnitude of the phenomenon in a given population, resulting from the actual number of deaths and the actual size of the population, standardised rates describe a certain hypothetical situation in which the study population would adopt the structure of a certain conventional reference population (in this study, the European Standard Population of 2013) [22].

Women in Poland live longer and outnumber men in older age groups, in which the prevalence of and mortality due to COPD are higher [23]. The greater rate of reduction in standardised mortality rates compared to crude ones was related to the lack of an effect of age on their value and the elimination of the association between the change in the proportion between age groups in favour of the elderly and the increase in mortality due to COPD [24]. The significant decrease in standardised mortality rates among women of 1.8% per year, with a concomitant stagnation in the values of crude rates in this population, means that the favourable trend of decreasing mortality by age group in this population was offset by an increase in the number of age groups with higher mortality from COPD. Similarly, in the male population, the ratio of reduction in standardised rates was greater than that of crude rates (5.6% per year vs. 3.4% per year), but, in their case, the positive trend of mortality decline outweighed the trend of unfavourable change in the age structure.

To sum up, mortality due to COPD was more common among males than females and, although the period of 2008–2021 saw a significant reduction in this disparity, it is noteworthy that the excess mortality determined based on standardised rates is significantly higher than that based on the crude rates. This means that the absolute magnitude of this phenomenon is smaller due to differences in the age structure in both sexes caused by the shorter life expectancy of males.

### 4.2. Comparison with Other Studies

Male excess mortality from COPD is common worldwide. In 2019, the excess mortality rate for men due to COPD, described by the ratio of male and female standardised rates, was 1.68 globally. The excess mortality rate level of 2.4 in Poland in 2021 was high; yet, it was comparable to the level for Central European countries (2.5). Higher values of excess mortality rates were recorded in 2019 only in Eastern European and high-income Asia Pacific countries and were 3.4 and 2.85, respectively [25]. The decline in mortality rates due to COPD is a worldwide phenomenon as well. However, only in wealthy countries, characterised by a high level of social development, is a reduction in male excess mortality observed as a result of a reduction in male mortality at a greater rate than a reduction in female mortality. In less developed countries, there is a greater reduction in female mortality than in male mortality [25]. 

A study conducted for EU countries and the UK showed that the decline in male excess mortality due to COPD is widespread in these countries. In Sweden and the UK, the incidence of COPD, but not the mortality, among women has been higher than among men in recent years [26]. The reduction in excess male mortality caused by COPD resulting from a significant decrease in mortality in men and a minor decrease in deaths in the female population is the most commonly observed pattern in European countries. COPD mortality is decreasing in the male population in 25 out of 28 European countries, including EU members and the UK, while mortality in the female population is decreasing in 19 out of 28 countries [26]. It should be noted, however, that while a decrease in the male mortality reduction rate has been described in many European countries in recent years, this Joinpoint analysis has shown a consistently high level of male mortality reduction in Poland over the entire study period (2008–2021), amounting to 5.6% per year for standardised rates [26]. Analyses carried out in the USA and Canada also indicate decreasing male excess mortality due to COPD. However, this is due to decreasing male mortality with relatively stable female mortality levels [27,28]. 

The reduction in male excess COPD-related mortality is of a similar magnitude to the reduction in excess mortality attributable to lung malignancies in Poland, which stood at 4.1 per cent per year between 2002 and 2017. While the decrease in male excess mortality due to COPD is a consequence of a simultaneous decrease in mortality in the male and female population, the decrease in excess mortality due to lung malignancies results from a simultaneous increase in female mortality accompanied by a slight decrease in mortality in the male population [29].

It should be noted that this paper is a descriptive epidemiological study. It aims to describe trends in excess mortality in men due to COPD in Poland. The methodology used does not allow the authors to demonstrate a cause–effect relationship between changes in exposure to COPD risk factors and COPD mortality. Furthermore, the authors of this study have only gender-specific data on exposure to tobacco smoke. This study’s research value would undoubtedly have been enhanced by the inclusion of potential differences in change in exposure to other relevant COPD risk factors and this certainly merits further exploration.Another limitation of this study is the potential underestimation of the number of deaths due to COPD in Poland. This disease is usually accompanied by other comorbidities, such as cardiovascular disease or cancer, which significantly affect death statistics. A further potential reason for the underestimation of COPD-related mortality is the mediocre quality of data on causes of death specified in death certificates. It is estimated that the cause of one in four deaths in Poland is determined using a so-called ‘garbage code’, making it impossible to identify the disease responsible for death accurately.

## 5. Conclusions

Mortality due to COPD is more common in the male population than in the female population, although the period of2008–2021 brought a significant reduction in this disproportion in Poland. The reduction in the level of male excess mortality in Poland was due to a simultaneous decrease in mortality in the male and female populations; yet, the decrease was greater in the male population.

Attention is drawn to the greater dynamics of reduction in the values of standardised rates than in the crude ones, both in the male and female populations, as well as the greater dynamics of decrease in the values of male excess mortality rates determined as a quotient of standardised rates. This is due to the elimination of the effect of changes in the ratio between age groups in favour of the elderly on changes in the level of mortality caused by COPD.

It should be noted that the mortality of men and women, as well as the excess mortality of men caused by COPD in Poland, over the study period declined more rapidly than in other European countries.

## Figures and Tables

**Figure 1 healthcare-12-00437-f001:**
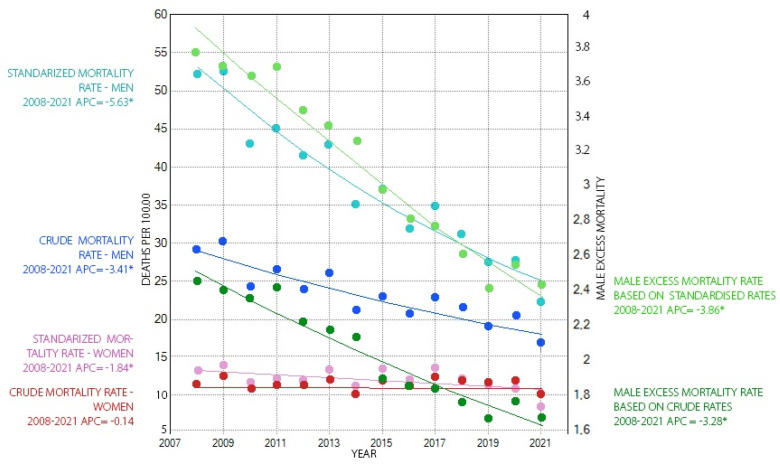
Mortality rates due to COPD by gender in the years 2008–2021 (left ordinate axis) and male excess mortality from COPD in the years 2008–2021 (right ordinate axis). * significant values were marked at *p* < 0.05.

**Table 1 healthcare-12-00437-t001:** Mortality rates due to COPD by gender per 100,000 (crude and standardised mortality rates).

Gender	Year													
	2008	2009	2010	2011	2012	2013	2014	2015	2016	2017	2018	2019	2020	2021
Women														
Crude Mortality Rates	12.0	12.5	10.5	11.0	11.1	11.9	10.1	12.2	11.2	12.5	12.3	11.9	11.6	10.1
Standardised Mortality Rates	13.7	14.2	11.8	12.2	12.1	12.8	10.7	12.8	11.4	12.6	12.1	11.4	10.9	9.2
Men														
Crude Mortality Rates	29.4	30.1	24.8	26.5	24.5	25.8	21.5	23.1	20.8	23.0	21.6	19.7	20.5	16.9
Standardised Mortality Rates	52.2	52.6	43.1	45.2	41.5	42.9	35.1	37.1	32.0	34.9	31.2	27.5	27.8	22.4

**Table 2 healthcare-12-00437-t002:** Percentage of people who smoke tobacco every day—2009–2017.

Gender	Year					
	2009	2011	2013	2015	2017	2019
Men	35%	39%	31%	31%	29%	24%
Women	24%	23%	23%	18%	20%	18%

## Data Availability

Data are available in a publicly accessible repository that does not issue DOIs. Publicly available datasets were analysed in this study. These data can be found here https://demografia.stat.gov.pl/bazademografia/Tables.aspx accessed on 10 August 2023.

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
