# Peer review of "Excess Mortality of Males Due to Chronic Obstructive Pulmonary Disease (COPD) in Poland"

_healthcare, 2024, doi:10.3390/healthcare12040437_

Round 1

Reviewer 1 Report

Comments and Suggestions for Authors

Dear authors,

I read your manuscript with great interest. Below you will find some doubts and suggestions that I hope can contribute to the final version.

Line 61-63: I don´t understand how this (as well as the and the begining of introduction)relates to the title. The mortality is decreasing, right? I know this becomes clear as you read on, but I read the entire introduction wondering what you were really going to study

Linha 170-172: As far as I understood, variables such as atmospheric pollution, workplace exposure and exposure to biomass combustion have not been considered and may also explain disparities. Maybe is what you mean in line 173. Could this be considered a study limitation?

 Line 252: The authors did not come across any limitations that may have, in any way, influenced the results. Also, wouldn't it be interesting to reflect on what can be further studied to understand these results even better?

Kind regards

Comments on the Quality of English Language

I think that only some review of the punctuation is necessary.

Author Response

Dear Sir/Madam

Thank you very much for your review.

I consider your remarks very valuable. I have analysed them with great diligence and modified the content of the manuscript to implement them. I believe that thanks to the advice provided, the work becomes clearer and more valuable for the readers.

Please find my detailed replies to the individual passages of the article you have indicated. The changes were also introduced in the paper.

1

Line 61-63: I don´t understand how this (as well as the and the begining of introduction)relates to the title. The mortality is decreasing, right? I know this becomes clear as you read on, but I read the entire introduction wondering what you were really going to study

The beginning of the introduction has been rewritten. We now inform the reader at the outset what the subject of the study is and what the results of our analysis are, i.e., the decline in male excess mortality.

2

Linha 170-172: As far as I understood, variables such as atmospheric pollution, workplace exposure and exposure to biomass combustion have not been considered and may also explain disparities. Maybe is what you mean in line 173. Could this be considered a study limitation?

In our work, we do not analyse the changes in the exposure of specific groups to the above factors. We have included an overview of exposure trends among men and women to smoking only. It would certainly provide a lot of important information if more detailed data on exposure to other important factors in the development of COPD by age group and gender were available. Unfortunately, the authors of this publication do not have such data, which is a certain constraint on our work.

3 

Line 252: The authors did not come across any limitations that may have, in any way, influenced the results. Also, wouldn't it be interesting to reflect on what can be further studied to understand these results even better?

A paragraph has been added on the limitations of our work and areas for further research.

It should be noted that this paper is a descriptive epidemiological study. It aims to describe trends in excess mortality in males due to COPD in Poland. The methodology used does not allow us to demonstrate a cause-effect relationship between changes in exposure to COPD risk factors and COPD mortality. Furthermore, only the data on exposure to tobacco smoke by gender is available to the authors of this study. Including potential differences in the change in exposure to other relevant risk factors for COPD in the study would undoubtedly increase the scientific value of this publication and certainly merits further investigation. Another limitation of this study is the potential underestimation of the number of deaths due to COPD in Poland. The disease is usually accompanied by other comorbidities, such as cardiovascular disease or cancer, which significantly affect death statistics. A further potential reason for the underestimation of COPD-related mortality is the mediocre quality of data on causes of death identified in death certificates. It is estimated that one in four deaths in Poland is described by means of a so-called ' garbage code', which makes it impossible to accurately identify the condition responsible for death.

Reviewer 2 Report

Comments and Suggestions for Authors

Dear Editor,

Please find below my opinions about the article you sent for my review.

Healthcare journal is an academic journal where articles of high novelty and quality are published. Although it is a topic of epidemiological importance, I do not think that the article titled "Excess Mortality of Males due to Chronic Obstructive Pulmo-2 nary Disease (COPD) in Poland." has sufficient original value and the necessary methodological competence.

With regret I must reject to publish this article in Healthcare journal.

Yours sincerely

Author Response

Dear Sir/Madam

Thank you very much for your review.

Below I present the answers in detail to the individual sections of the article you have indicated. Changes have also been made to the manuscript.

1.

The introduction to the article has been rewritten. We now inform the reader at the very beginning what constitutes the subject of the study, and what its result is.

2.

The study has been supplemented with a review of trends in smoking habits among males and females in Poland. A comparative table in this regard has also been added.

3.

A paragraph on the limitations of our work and areas for further research has been added. We have pointed out that the only figures available are those on exposure to tobacco smoke by gender. Should potential differences in the change in the exposure to other relevant risk factors for COPD be included in the study, its scientific value would undoubtedly be enhanced. This certainly deserves further investigation.

Reviewer 3 Report

Comments and Suggestions for Authors

A retrospective descriptive study on the trend of COPD mortality in Poland presents a fundamental bias: when it indicates that the decrease in excess COPD mortality is greater in men, this is something well known in other countries, particularly in Europe. This steep decline is probably due to the decline in smoking in the male population (or not) and nothing is said about it in the study. I believe that the limitations of the study should address this issue although the most desirable would be to review the trends in smoking habits in Polish men and women to be able to draw more extrapolated conclusions about mortality declines. A comparative table in this regard is lacking. I believe that this is the great weakness of the study and should be corrected.

Author Response

Dear Sir/Madam

Thank you very much for your review.

I consider your remarks very valuable. I believe that thanks to the advice provided, the work becomes clearer and more valuable to the readers.

The study has been supplemented with a review of trends in smoking habits among men and women in Poland. A comparative table in this regard has also been added.

Round 2

Reviewer 2 Report

Comments and Suggestions for Authors

Dear Authors,

Please find below my comments on the changes made by the authors in the article titled "Excess Mortality of males due to Chronic Obstructive Pulmonary Disease (COPD) in Poland." that you sent for my review. Although the introduction, method and findings sections have been revised and further analyses and improvements have been made, I believe that the data provided does not have sufficient scientific rigour.

I regret that the analysis and conclusions made in this study, in which the novelty is one of the major problems, are not at an sufficient level, and I regret that this article is not suitable to be published in Healthcare journal.

Yours sincerely

Author Response

Dear Sir/Madam

Thank you very much for your review.

We have made the following changes in response to your suggestions for improving our work.

  1. The introduction has been supplemented to provide a broader background and include more relevant references.
  2. The references have been reassessed and more relevant references have been added
  3. The Results have been rewritten, so that the presentation of the results is more transparent
  4. The Results have been rewritten, making the relationship between results and conclusions more conspicuous.

Yours sincerely

Reviewer 3 Report

Comments and Suggestions for Authors

With the modifications made, it can be accepted for publication.

Author Response

Dear Sir/Madam

Thank you very much for your review.